# Effect of Caffeine Supplementation on Sports Performance Based on Differences Between Sexes: A Systematic Review

**DOI:** 10.3390/nu11102313

**Published:** 2019-09-30

**Authors:** Juan Mielgo-Ayuso, Diego Marques-Jiménez, Ignacio Refoyo, Juan Del Coso, Patxi León-Guereño, Julio Calleja-González

**Affiliations:** 1Department of Biochemistry, Molecular Biology and Physiology, Faculty of Health Sciences, University of Valladolid, 42004 Soria, Spain; 2Academy Department, Deportivo Alavés SAD, 01007 Vitoria-Gasteiz, Spain; dmarques001@ikasle.ehu.eus; 3Department of Sports, Faculty of Physical Activity and Sports Sciences (INEF), Universidad Politécnica de Madrid, 28040 Madrid, Spain; ignacio.refoyo@upm.es; 4Centre for Sport Studies. Rey Juan Carlos University, 28943 Fuenlabrada, Spain; juandelcosogarrigos@gmail.com; 5Faculty of Psychology and Education, University of Deusto, Campus of Donostia-San Sebastián, 20012 San Sebastián, Guipúzcoa, Spain; patxi.leon@deusto.es; 6Laboratory of Human Performance, Department of Physical Education and Sport, Faculty of Education, Sports Section, University of the Basque Country, 01007 Vitoria, Spain; julio.calleja.gonzalez@gmail.com

**Keywords:** recovery, strength, power, sprint performance, menstrual cycle

## Abstract

Most studies that have shown the positive effects of caffeine supplementation on sports performance have been carried out on men. However, the differences between sexes are evident in terms of body size, body composition, and hormonal functioning, which might cause different outcomes on performance for the same dosage of caffeine intake in men vs. women. The main aim of this systematic review was to analyze and compare the effects of caffeine intake between men and women on sports performance to provide a source of knowledge to sports practitioners and coaches, especially for those working with women athletes, on the use of caffeine as an ergogenic aid. A structured search was carried out following the Preferred Reporting Items for Systematic Review and Meta-Analyses (PRISMA) guidelines in the Web of Science, Cochrane Library, and Scopus databases until 28 July 2019. The search included studies in which the effects of caffeine supplementation on athletic performance were compared between sexes and to an identical placebo situation (dose, duration and timing). No filters were applied for participants’ physical fitness level or age. A total of 254 articles were obtained in the initial search. When applying the inclusion and exclusion criteria, the final sample was 10 articles. The systematic review concluded that four investigations (100% of the number of investigations on this topic) had not found differences between sexes in terms of caffeine supplementation on aerobic performance and 3/3 (100%) on the fatigue index. However, four out of seven articles (57.1%) showed that the ergogenicity of caffeine for anaerobic performance was higher in men than women. In particular, it seems that men are able to produce more power, greater total weight lifted and more speed with the same dose of caffeine than women. In summary, caffeine supplementation produced a similar ergogenic benefit for aerobic performance and the fatigue index in men and women athletes. Nevertheless, the effects of caffeine to produce more power, total weight lifted and to improve sprint performance with respect to a placebo was higher in men than women athletes despite the same dose of caffeine being administered. Thus, the ergogenic effect of acute caffeine intake on anaerobic performance might be higher in men than in women.

## 1. Introduction

Numerous studies have shown the effectiveness of caffeine supplementation on sports performance in which aerobic [1], anaerobic [2,3,4] or mixed [5,6,7] metabolism is prioritized. Current guidelines recommend the ingestion of low-to-moderate doses of caffeine, ranging from 3 to 6 mg/kg, approximately 60 min prior to exercise to get these improvements [8,9]. Higher doses of caffeine (9–13 mg/kg) do not result in an additional improvement in physical performance [10], while these higher doses might increase the incidence and magnitude of main caffeine-related side effects. In addition, high doses of caffeine might end in urine caffeine concentrations greater than 15 µg/ml, which is prohibited in the National Collegiate Athletic Association (NCAA) [11].

In general, several mechanisms have been proposed to explain the effects of caffeine supplementation on sports performance [3,12]. However, the most well-recognized mechanism at present is that caffeine acts in the central nervous system (CNS) as a competitor for adenosine in its receptors, inhibiting the negative effects that adenosine induces on neurotransmission, excitation and pain perception [13]. In addition, the hypoalgesic effect of caffeine decreases the perception of pain and effort during exercise and therefore might also be considered as a supplementary mechanism of action, at least for exercise situations that induce pain [3,12]. As a result, lower pain perception could maintain or increase the firing rates of the motor units and possibly produce a more sustainable and forceful muscle contraction, and consequently, allow greater strength production [3,14].

Caffeine can affect the use of energy substrates during exercise. In particular, it has been suggested that caffeine supplementation acts as a glycogen saver as it increases the mobilization of free fatty acids by adrenaline (epinephrine) induction [15]. Although this mechanism could favor aerobic and anaerobic sports that depend on muscle glycogen, it is currently known that there are other mechanisms by which athletic performance would be favored such as increased calcium mobilization and phosphodiesterase inhibition [3,9]. In addition, it has been proposed that caffeine supplementation causes a greater activity of the Na^+^/K^+^ pump to enhance excitation contraction coupling [16].

Given that sex has been identified as an important determinant of athletic performance through the impact of body composition, aerobic capacity or anaerobic thresholds due to hormonal differences [17], specific recommendations for each sex should be in agreement with these sex differences to achieve better results in sports performance. In this respect, while there is a position and recommendations about the use of caffeine supplements in athletes [18,19], there is not enough comparative information on the effects of caffeine on athletic performance between men and women athletes [20]. For that reason, caution would be needed in extrapolating the recommendations made for men to women, since the vast majority of the studies included only male participants [1,2,5,21]. In fact, only ~13% of participants in investigations aimed to determine the ergogenic effect of caffeine are women, while the effect of caffeine in women at high (>9 mg/kg) or very low doses (<1 mg/kg) is unexplored [22]. In addition, due to the menstrual cycle, women are subject to hormonal changes that could affect sports performance [23,24]. For instance, it has been shown that the phase of the menstrual cycle influences the development of strength [25]. Also, the consumption of oral contraceptives has effects on the metabolism of caffeine, extending its half-life and prolonging the responses in the human body [26], although very few studies take these aspects into account [27]. 

In this respect, studies conducted with the general population have already shown that the stimulating effects (less drowsiness and greater activation) of caffeine are greater in men than in women [28]. Still, few studies have shown the differences in the effect of caffeine supplementation on sports performance between men and women, and their results are controversial [29,30,31,32,33,34,35,36,37]. While some studies have shown a comparable ergogenic effect of caffeine between sexes on sports performance, others have presented a greater effectiveness of caffeine to increase sprint power [29], isolated forehand stroke peak and average speed [37], total weight lifted [38] and a shorter time to perform a repeated modified agility test (RMAT) [30] in men compared to women. Unifying the data from these different studies could provide knowledge regarding the effect that caffeine supplementation has on sports performance based on the athlete’s sex. This analysis might help to enhance the recommendations for caffeine supplementation based on sex and the type of exercise performed. Therefore, it was proposed to carry out a systematic review of the relevant articles published in the scientific literature.The main objective of which was to discern the possible effects of caffeine supplementation on sports performance based on the participant’s sex. Specifically, this systematic review focuses on determining the different responses between the sexes to the same caffeine supplementation protocol depending on whether the exercise will be classified as aerobic, anaerobic or when the protocol induced some type of fatigue that could be evaluated (i.e., index of fatigue).

## 2. Methodology

### 2.1. Search Strategy

This article is a systematic review focused on the performance effects of caffeine in men athletes vs. women athletes. It was carried out following the Preferred Reporting Items for Systematic Review and Meta-Analyses (PRISMA) guidelines [39]. A structured search was carried out in the Web of Science (WOS), which includes other databases such as BCI, BIOSIS, CCC, DIIDW, INSPEC, KJD, MEDLINE, RSCI, SCIELO, and the Cochrane Library and Scopus, sources of high-quality information in the field of health sciences, thus guaranteeing complete bibliographic support. The search strategy ended on 28 July 2019. The search terms included a mix of medical subject headings (MeSH) and free-text words for key concepts related to caffeine, the sex of the athletes under investigation and different forms of exercise and sports performance. The following search equation was used to find the relevant articles: *(“caffeine”[MeSH Terms] OR “caffeine”[All Fields]) AND ((“female”[MeSH Terms] OR “female”[All Fields]) OR (“women”[MeSH Terms] OR “women”[All Fields])) ((“male”[MeSH Terms] OR “male”[All Fields]) OR (“men”[MeSH Terms] OR “men”[All Fields] OR “woman”[All Fields])) AND ((“exercise”[MeSH Terms] OR “exercise”[All Fields]) OR (“sports”[MeSH Terms] OR “sports”[All Fields] OR “sport”[All Fields])) AND performance [All Fields].* No filters were applied to the athlete’s physical fitness level, race, or age to increase the power of the analysis. The search for published studies was independently performed by 2 different authors (JMA and JCG). 

### 2.2. Inclusion and Exclusion Criteria 

The PICOS model was used to determine the inclusion criteria [40]: P (Population): “men and women athletes”, I (Intervention): “caffeine supplementation”, C (Comparators): “identical conditions for caffeine and placebo experimental trials”, O (Outcome): “physical and/or sports performance measurements”, and S (study design): “single- or double-blind and randomized design”.

As a result, the studies included in this systematic bibliographic review had to meet all the following criteria: (i) populations were elite or amateur athletes or active people, men and women of any age; (ii) participants performed any form of physical exercise or sport using caffeine as an ergogenic aid, which could be administered in the form of capsules/pills, energy or sports drinks, commercial drinks with caffeine content, chewing gum or coffee; (iii) the effects of caffeine were compared on both sexes to an identical placebo condition and the protocols used were similar for male and female participants; (iv) articles examined the effects of caffeine supplementation on physical performance measurements, physiological responses, perceptual measures; (v) study designs were randomized, single- or double-blind, and placebo-controlled. The following exclusion criteria were applied to the experimental protocols of the investigation: (i) studies that were conducted only in men or in women athletes; (ii) studies that were performed for clinical purposes or therapeutic use; (iii) the absence of a true placebo condition or different experimental protocols used for male and female participants; (iv) studies carried out using participants with a previous cardiovascular, metabolic, or musculoskeletal disorder.

### 2.3. Study Selection

Two authors identified papers through a database search (JMA and JCG). The titles and abstracts of publications identified by the search strategy were screened for a subsequent full-text review and were cross-referenced to identify duplicates. All trials assessed for eligibility and classified as relevant were retrieved and the full text was peer-reviewed (JMA and JCG). Moreover, the reference sections of all relevant articles were also examined applying the snowball strategy. Based on the information within the full reports, inclusion and exclusion criteria were used to select the studies eligible for inclusion in the systematic review. Disagreements were resolved through discussions between the different authors (JMA and JCG).

### 2.4. Data Extraction

Once the inclusion/exclusion criteria were applied to each study, the following data were extracted: study source (author/s and year of publication); population of the sample indicating the level of activity or sports discipline, age, sex and number of participants; habitual caffeine intake (mg/day); dose of caffeine intake, source from which it is obtained and its administration protocol; and performance outcomes in men and women.

### 2.5. Quality Assessment and Risk of Bias 

In order to carefully consider the potential limitations of the included studies to obtain reliable conclusions, and following Cochrane Collaboration Guidelines [41], two authors independently assessed the methodological quality and risk of bias (JMA and DMJ) of each investigation, and disagreements were resolved by third-party evaluation (JCG). In the Cochrane Risk of Bias tool, the following items are included and divided into different domains: (1) selection bias (items: random sequence generation, allocation concealment), (2) performance bias (blinding of participants and personnel), (3) detection bias (blinding of outcome assessment), (4) attrition bias (incomplete outcome data), (5) reporting bias (selective reporting), and (6) other bias (other sources of bias). The assessment of the risk of bias was characterized as low risk (plausible bias unlikely to seriously alter the results), unclear risk (plausible bias that raises some doubt about the results), or high risk (plausible bias that seriously weakens confidence in the results).

## 3. Results

### 3.1. Search Strategy

After applying the search equation, a total of 202 records were identified through database searches and six studies through reference list searches. From these 208 articles, 45 of them were removed because they were duplicates. In addition, 33 studies were excluded after screening the abstract. As a result, 129 studies were assessed for eligibility. From the 129 full-text articles assessed, another 119 papers were removed because they were unrelated to the topic of this systematic review. The topics and number of studies that were excluded were as follows: those excluded because the subjects were inappropriate for the inclusion criteria (*n* = 51; in animals *n* = 10; and in the general population *n* = 41), those that used an unsuitable methodology (*n* = 33; outcomes in men and women separately, or did not compare the responses for both sexes), and those with unsuitable outcomes (*n* = 29; cognitive function and sleep *n* = 23; and toxicological and genetic studies *n* = 5; and bibliographic reviews *n* = 7). Consequently, 10 studies met the previously defined inclusion criteria and were included in this final systematic review (Figure 1).

### 3.2. Caffeine Supplementation

The total sample consisted of 221 participants (*n* = 113 males; *n* = 108 females) [29,30,31,32,33,34,35,36,37,38]. All studies were performed using adult populations. Healthy active students were selected in three studies [30,31,37] while the remaining studies included participants catalogued as athletes because they train for a specific sport. Athletes from endurance sports such as cycling, triathlon [29,32,36], and from resistance training modalities were used in the investigations [35,38]. Moreover, two trials were performed with elite collegiate athletes from several disciplines such as tennis, basketball and soccer [33,34].

The sources of caffeine supplementation were varied, including commercial drinks used by Jacobson et al. (2018) [37], Tinsley et al. (2017) [35], high chlorogenic coffee (Turkish coffee) used by Nieman et al. (2017) [32], dry anhydrous caffeine mixed with 300 mL water and a sugar-free peach squash solution proposed by Sabblah et al. (2015) [38], and caffeine gum used by Paton et al. (2015) [29]. The other authors of this systematic review used capsules to administer the scheduled doses of caffeine in their studies [31,33,34,36].

In the included studies, caffeine was administered in different doses, based on an individual’s body mass, or with an absolute dose. The doses based on the participant’s body mass used between 3 and 6 mg/kg of body mass. Skinner et al. (2019) used 3 mg/kg [36], Paton et al. (2015) used 3–4 mg/kg [29], and Tinsley et al. (2017) provided 4 mg/kg of caffeine for men and 3.6 mg/kg of caffeine for women using a caffeinated supplement [35]. Besides, Jebabli et al. (2016) [30] and Sabblah et al. (2015) [38] used 5 mg/kg in the caffeine administration protocol, while Chen et al. (2015) [33], Chen et al., (2019) [34], and Suvi et al. (2016) [31] used a dose of 6 mg/kg. In relation to studies that provided an absolute dose, participants in the study by Jacobson et al. (2018) [37] consumed a commercially available energy drink with 240 mg (≈3.1 mg/kg) of caffeine, and Nieman et al. (2017) [32] used 474 mg (men: ≈6.7 mg/kg and women: ≈7.5 mg/kg) of caffeine from a cup of coffee in their study.

In general, the time of the ingestion of caffeine was between 30 and 60 min before testing. Thus, Chen et al. (2015) [33], Chen et al. (2019) [34] and Sabblah et al. (2015) [38] agreed on administering this type of supplementation 60 min before testing, and Tinsley et al. (2017) [35] and Jacobson et al. (2018) [37] prescribed caffeine supplementation 30 min before testing. However, 45 min before [30] and 90 min before [36] testing were also selected as time-points of caffeine supplementation. Likewise, Suvi et al. (2016) [31] fractionated the dose into two portions: 60 min before (4 mg/kg) and immediately prior to testing (2 mg/kg). A different strategy for supplementation was chosen by Nieman et al. (2017) [32], who proposed a protocol of chronic intake of Turkish coffee every morning for two weeks. In contrast, only one study [29] administered caffeine supplementation during exercise (after completing one third of a 30 km test), and another one after exercise [34], where participants ingested caffeine at 24 and 48 h post-exercise.

### 3.3. Outcome Variables 

Studies included in this systematic review measured a large range of variables. Consequently, studies were clustered by the character of the measurements, such as aerobic performance (Table 1), anaerobic performance (Table 2) and the fatigue index (Table 3). As a result, the effects of caffeine supplementation on aerobic performance were analyzed in four studies [29,31,32,36], on anaerobic performance in seven studies [29,30,31,32,33,34,35,36,37,38] and on the fatigue index in three studies [30,33,34]. 

### 3.4. Quality Assessment and Risk of Bias

In relation to selection bias, random sequence generation was characterized as low risk only in three studies [35,36,37], while in the remaining studies, the bias was unclear [29,30,31,32,33,34,38]. Allocation concealment was categorized as low risk in all experiments [29,30,31,32,33,34,35,36,37,38]. Regarding performance bias, the blinding of participants was categorized as low risk in nine studies [29,31,32,33,34,35,36,37,38] and high risk in one trial [30], whereas the blinding of personnel was categorized as low risk in four studies [33,34,35,36], unclear in five trials [29,30,31,32,37], and high risk in one trial [38]. The domain attrition bias, measured by incomplete outcome data, shows that six studies can be characterized as low risk [29,30,31,32,37,38], and four studies can be considered as unclear risk [33,34,35,36]. In relation to reporting bias, evaluated through selective reporting, five trials were considered to be of low risk [31,33,34,36,37], three to be of unclear risk [29,35,38] and two to be of high risk [30,32]. Finally, six studies were characterized as low risk of other bias [29,31,33,34,36,38], two trials as unclear risk [32,35] and two studies as high risk [30,37]. Full details for all these risks are given in Figure 2 and Figure 3.

## 4. Discussion

The main aim of this systematic review was to summarize the differences, if any, in the ergogenic effect of caffeine supplementation between men and women. This systematic review focuses on different responses between sexes to the same caffeine supplementation protocol when the exercise was categorized as aerobic, anaerobic or when the protocol induced some kind of fatigue that could be assessed (i.e., fatigue index). Knowing that caffeine is one of the most popular ergogenic aids with demonstrated effects on physical performance [6,18,19], it was considered that this systematic review might show a global vision of the ergogenic effect of caffeine in men and women, gathering all studies published in this field. Generally, the investigations analyzed reflect that there are no differences between men and women. However, some investigations showed some subtle differences between sexes, indicating that males might experience an increased ergogenic effect of caffeine, especially to produce more power, greater total weight lifted and higher speed with the same dose of caffeine. These results suggest that, in general, both men and women athletes benefit from caffeine supplementation to the same extent. Nevertheless, it seems that ergogenicity might be greater in men for exercise activities with an anaerobic component. Interestingly, none of the investigations depicted a negative effect of acute caffeine intake on physical performance. All this information indicates that the current guidelines for caffeine supplementation can be equally valid for men and women athletes.

### 4.1. Effects of Caffeine on Aerobic Performance

Caffeine has been popularly used in long-lasting sports given that some improvements are observed over time to exhaustion [8,9], mainly due to a hypothetical glycogen-sparing effect of caffeine [15] and the stimulation in the CNS of this substance, capable of attenuating pain [42]. In fact, numerous studies have used the duration of a test and the time to complete it as an indicator of aerobic endurance performance, in which the effect of a supplement such as caffeine can be measured [29,30,31,32,36].

Regarding the differences between men and women, there are more studies in which there are no significant differences between sexes in time trials. For example, Paton et al., (2015) [29] showed improvements of a similar magnitude in the aerobic performance of both sexes, but without differences between them. However, these authors found that men showed significant improvements in anaerobic performance with respect to women, probably due to the increase in activity of the CNS. One of the main reasons why they did not find significant differences between sexes could be the great interindividual variability, which could be associated with individual differences in the metabolism of caffeine or the absorption rate [43]. 

Suvi et al. (2016) [31] did not show significant differences between sexes in endurance capacity in hot environments (42 °C, 20% relative humidity) after taking 6 mg/kg of caffeine in two doses (4 mg/kg 60 min before the test and 2 mg/kg immediately before it). Similarly, Nieman et al. (2017) [32] did not find sex differences after chronic intake of 474 mg of caffeine for 2 weeks either. Moreover, Skinner et al. (2019) [36] showed significant and comparable improvements in endurance performance in both sexes. The performance improvements observed in women were similar to those in men, even though women showed higher plasma caffeine concentrations. Thus, Skinner et al. (2019) [36] suggested that current recommendations for caffeine supplementation, which are derived from studies conducted in men, could also be applied to women, in particular in aerobic endurance events. The results of these studies suggest that caffeine supplementation is equally effective in terms of improving the aerobic capacity in both men and women athletes.

### 4.2. Effects of Caffeine on Anaerobic Performance

Regarding anaerobic actions, it has been demonstrated that caffeine produces positive effects on performance among others due to the activation of the CNS [3,44]. Caffeine might antagonistically bind adenosine receptors and decrease adenosine-mediated fatigue [45]. Moreover, a recent study has also suggested that caffeine in physiological concentrations (~40 μmol/L) may improve calcium release from the sarcoplasmic reticulum during muscle contraction [46]. To the best of the authors’ knowledge, only one study conducted by Jababli et al. (2016) [30] compared the caffeine supplementation effect on the glycolytic pathway in both sexes. The authors observed that the effect of caffeine to reduce the total time employed to complete several repetitions of an agility test was higher in men than women. The authors indicated that one explanation could be the increase in alertness, leading to more concentration during the execution of this type of test, especially during the changes of direction [30]. However, the effect of caffeine was similar when only the best time in the agility test was taken into account. The controversies between these results may be due to either the sex selection or the selected study population. Due to the scarcity of data comparing the effect of caffeine on agility in men and women, more studies are necessary to confirm these results.

Anaerobic capacity manifests the rapid use of the phosphagen system (adenosine triphosphate (ATP) and creatine phosphate) in the muscles, with type II fibers providing the greatest contribution. Anaerobic capacity can be estimated by numerous laboratory and field tests such as anaerobic speed test, mean power output and peak power output [47]. In addition, there is evidence about the ergogenic effect of caffeine on anaerobic capacity [3,30,44], but the causes remain unclear [48]. It is now accepted that caffeine induces higher levels of Ca^2+^ and K^+^. The influx of Ca^2+^ from the sarcoplasmic reticulum favors the formation of cross bridges and therefore increases muscle power, whereas the serum increase in K^+^ causes increases in the Na^+^ / K^+^ ATPase activity, so that it can attenuate muscle fatigue [33,49].

Regarding the potential sex differences of caffeine supplementation on muscle power, Paton et al. (2015) [29] presented similar increases in mean power in the last 10 km between men and women after 3–4 mg/kg caffeine chewing gum supplementation during a 30 km continuous cycling exercise with varying intensity. However, the men showed greater sprint power induced by caffeine than the women, due to the inclusion of aerobic phases of exercise prior to the sprints performed in the test, as well as the inter-individual variability in the response to caffeine [8]; thus, posing a possible sex -based difference that affects high intensity anaerobic measurements [50].

Jacobson et al. (2018) [37] found that only women improved forehand stroke velocity after the intake of 240 mg of caffeine, while the caffeine-induced changes on vertical jump was similar between sexes. Opposite results were shown in the Warren et al. (2010) meta-analysis [48], where it is stated that caffeine has more impact on exercise that involves large muscles than small muscles, such as in the arms. These differences could be, (a) related to the dose of caffeine supplementation because it could be adequate to improve the performance of small muscle groups, but insufficient to improve the performance of muscle groups in the legs and (b) the caffeine amount was the same for all participants, so that the differences in body mass and body composition between sexes could cause the women to receive more caffeine than the men [37].

Therefore, based on the results included in the studies of this systematic review, there are differences between the manifestation of variables associated with muscle power and speed between men and women. However, given the small number of studies, it is difficult to attribute the causes. Jebabli et al. (2016) [30] indicated that a dose of 5 mg/kg of caffeine 45 min before a repeated agility test decreased the total time more in men than in women [30]. The authors attributed these results to the increase in neuromuscular activity that facilitates the neural transmission observed in the men, of which there is no evidence in the women [51].

Another physical capacity included in anaerobic performance may be strength. In this respect, strength is based on a combination of morphological and neural factors that include the cross-sectional area of the muscle and architecture, musculotendinous stiffness, motor unit recruitment, frequency coding, motor unit synchronization and neuromuscular inhibition [52]. However, the effects of caffeine supplementation on strength are not so clear. Some studies found an acute effect of caffeine on increasing strength, while others did not present a response to supplementation with caffeine [48,53]. In any case, few studies have analyzed the potential differences between sexes evaluating the variables associated with strength.

Chen et al. (2015) [33] and Chen et al. (2019) [34] showed that 6 mg/kg of caffeine supplementation improved the maximum voluntary isometric strength (measured by isometric contractions) of the knee extensors after 60 min of caffeine ingestion, and 24/48 h post-exercise, respectively, but without significant differences between sexes. According to the authors, one of the main factors that may have influenced the absence of sex differences was that the women’s sample was homogenized in relation to the menstrual cycle (all were in the early follicular phase) [33,34]. In this respect, although some investigators have reported an inotropic effect of estrogens on muscle because of a switching of muscle cross-bridges from low- to high-force generation [54,55], in general muscle strength does not appear to fluctuate significantly during an ovulatory menstrual cycle [47].

In the same line of research, Tinsley et al. (2017) [35] did not show significant differences between sexes in the production of strength after caffeine supplementation (men: 4.0 mg/kg and women: 3.6 mg/kg of caffeine). However, the effect size showed that the caffeine supplement contributed to small increases (not significant) in the men′s concentric strength (5–20%, *d* = 0.2–0.4 relative to placebo), but not in the women′s. In this study, the women′s menstrual cycle phase was not taken into account. However, women participants were given 75% of the supplement given to men participants to match the differences in body composition between both sexes [35], which could have impacted the slightly lower response to caffeine in the women. 

Sabblah et al. (2015) [38] found that 5 mg/kg of caffeine had positive effects on 1RM for both sexes trained in resistance. However, it showed that the women’s reaction was smaller than men’s, as evidenced by a tendency to improve total weight lifted for men with no such effect in this variable for women. Besides, the authors showed that the perception of pain in both sexes revealed no differences after caffeine supplementation. The greater activation that caffeine has shown in men compared to women could be the potential cause of these differences in the total weight lifted [28].

Therefore, although there are equivocal results, there are more than reasonable doubts to consider that the effect of caffeine supplementation influences men and women differently as regards strength.

### 4.3. Effects of Caffeine on the Fatigue Index

The consumption of pre-workout supplements makes it possible to experience physiological effects, as well as the psychological effects on performance [35]. In this case, caffeine may delay the onset of fatigue or block the perception of pain or fatigue, using the same mechanisms involved in perception variables [33]. However, regarding rated perceived exertion there is scarcely any scientific evidence that could show significant differences between sexes. Three of the studies included in this systematic review do not show significant differences between men and women in the fatigue index [30,33,34,35]. Only the study published by Suvi et al. (2016) [31] showed that caffeine supplementation reduced the perception of fatigue in men, but not in women, when exercising in hot environments. The authors attributed this result to the greater sensitivity of men to acute caffeine ingestion compared to women, as shown by previous research [28,56,57,58].

## 5. Conclusions

In summary, 10 studies met the previously defined inclusion/exclusion criteria and were included in this systematic review, aimed to analyze the between-sex differences in the effect of caffeine supplementation on physical/sports performance. The total sample consisted of 221 participants (*n* = 113 males; *n* = 108 females), while caffeine supplementation was given by using different sources of caffeine between 30 and 60 min before testing. Studies included in this systematic review measured a large range of variables such as aerobic and anaerobic performance and the fatigue index. Overall, the caffeine supplementation produced a similar ergogenic benefit for aerobic performance and the fatigue index in men and women athletes. However, the effects of caffeine to produce more power, total weight lifted and to improve sprint performance with respect to a placebo was greater in men than women athletes despite the same dose of caffeine being administered. Specifically, the men experienced greater mean power than the women during the final 10-km cycling test after ~3–4 mg/kg caffeine supplementation. Likewise, the ergogenic effect of 5 mg/kg of caffeine to increase the total weight lifted was higher in men than in women, in particular the ergogenic effect of acute caffeine intake on anaerobic performance.

### Strengths, Limitations and Future Lines of Research

The main strength of the present study is its novelty, given that no previous systematic review has analyzed the effect of caffeine supplementation on performance between sexes. The main limitations were the scarcity of information and the low sample sizes used in most investigations. Besides, differences in physical tests and caffeine supplementation protocols make it difficult to generalize the recommendations. Finally, it is of great importance to continue the work in future research on supplementation with caffeine in female populations. Specifically, it is necessary to determine whether women athletes benefit from acute caffeine intake in other forms of exercise or in sports where several physical fitness variables affect overall performance. In addition, the between-sex difference in the response to other dietary supplements should also be investigated to ascertain whether the findings of previous investigations with males are also applicable to women athletes. Lastly, it is necessary to take into account women’s menstrual cycle when investigating caffeine ergogenicity to determine whether this substance exerts positive performance effects in all phases of the menstrual cycle.

## Figures and Tables

**Figure 1 nutrients-11-02313-f001:**
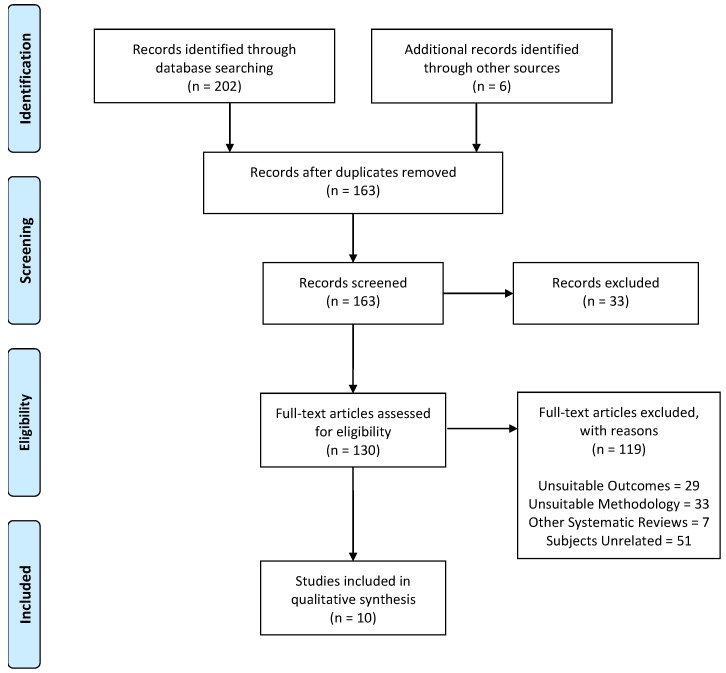
Selection of studies (Preferred Reporting Items for Systematic Review and Meta-Analyses (PRISMA), 2009 Flow Diagram).

**Figure 2 nutrients-11-02313-f002:**
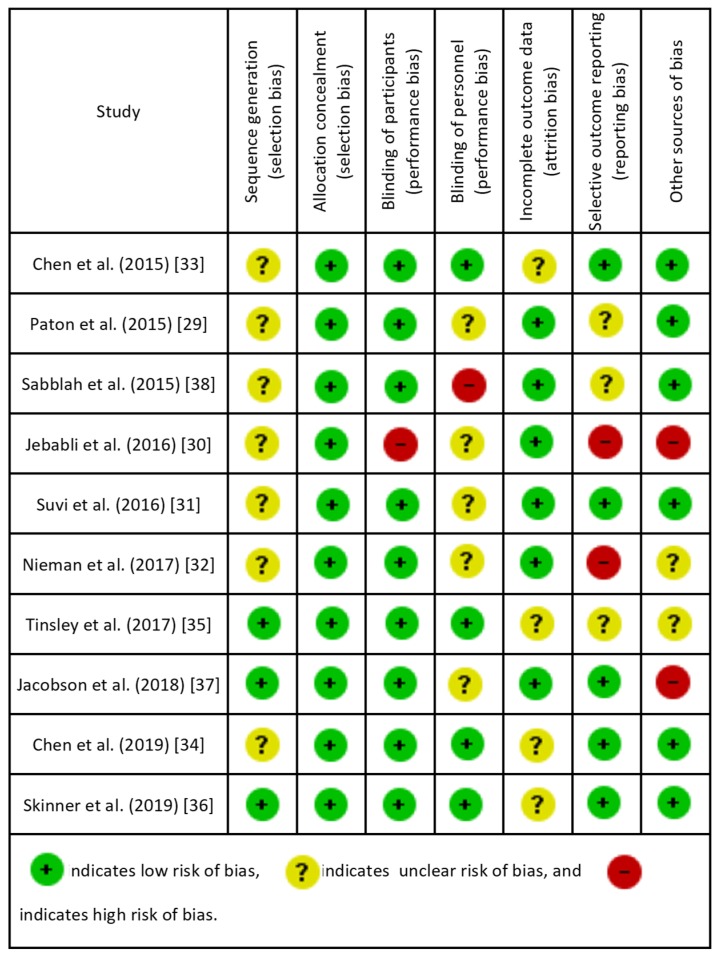
Risk of bias summary: review of authors´ judgements about each risk of bias item presented as percentages across all included studies.

**Figure 3 nutrients-11-02313-f003:**
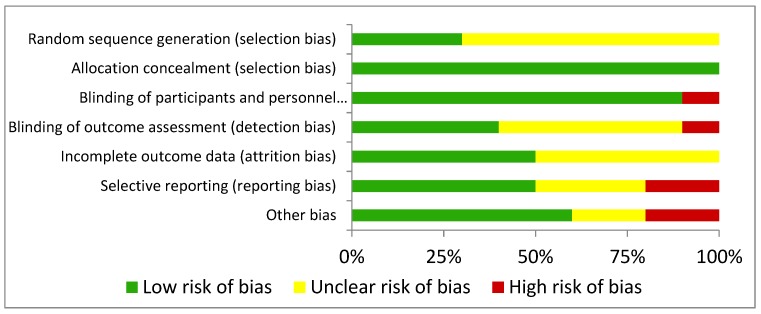
Risk of bias graph: review authors’ judgements about each risk of bias item for each included study.

**Table 1 nutrients-11-02313-t001:** Summary of studies included in the systematic review that investigated the effect of caffeine ingestion between sexes on aerobic performance.

Author/s	Population	Intervention	Main Outcome Analyzed	Effect on Men vs. Women
Paton et al. (2015) [29]	Trained cyclists 10 Men (36 ± 10 years) 10 Women (25 ± 7 years)	3–4 mg/kg in caffeinated gum During exercise (10-km point at 1st sprint)	• Time trial performance	• ↔ • ↔ • ↔
Suvi et al. (2016) [31]	Healthy active students 13 Men (24.9 ± 4.1) 10 Women (22.5 ± 2.0 years)	6 mg/kg of gelatin in capsule in two doses 4 mg/kg 60 min before 2 mg/kg pre-test	• Time to exhaustion (minute)	• ↔
Nieman et al. (2017) [32]	Cyclists 10 Men (36.1 ± 3.3) Five Women (40.0 ± 4.5 years)	474 mg of Turkish coffee (men: ≈6.7 mg/kg and women: ≈7.5 mg/kg) Each morning for two weeks	• Time trial performance	• ↔
Skinner et al. (2019) [36]	Endurance-trained cyclists and triathletes 16 Men (32.6 ± 8.3) 11 Women (29.7 ± 5.3 years)	3 mg/kg in opaque capsules 90 min before	• Time trial performance	• ↔

↔ The effect of caffeine supplementation was not statistically different between sexes.

**Table 2 nutrients-11-02313-t002:** Summary of studies included in the systematic review that investigated the effect of caffeine ingestion between sexes on anaerobic performance.

Author/s	Population	Intervention	Main Outcome Analyzed	Effect on Men vs. Women
Chen et al. (2015) [33]	Elite collegiate athletes (tennis, basketball, soccer) 10 Men (20.10 ± 2.18 years) 10 Women (19.9 ± 0.99 years)	6 mg/kg in capsules taken with 500 mL water 60 min before	• MVIC (Nm/kg) • SVIFP (s) • MVIC (Nm/kg)	• ↔ • ↔ • ↔
Paton et al. (2015) [29]	Trained cyclists 10 Men (36 ± 10 years) 10 Women (25 ± 7 years)	3–4 mg/kg in caffeinated gum During exercise (10-km point at 1st sprint)	• 0.2-km sprints each 10-km of a 30-km cycling time trial	• ↑ in men
Sabblah et al. (2015) [38]	Moderately active resistance-trained individuals 10 Men (24.4 ± 3.2 years) Eight Women (27.9 ± 6.13)	5 mg/kg of dry anhydrous caffeine mixed with 300 mL water and a sugar- free peach squash solution 60 min before	• Bench press 1RM • Squat 1RM • Number of bench press reps to failure at 40% 1RM (total weight lifted)	• ↔ • ↔ • ↑ in men
Jebabli et al. (2016) [30]	Healthy active students of Sports Sciences 10 Men (22.9 ± 1.46 years) Eight Women (21.8 ± 0.45 years)	5 mg/kg (Undefined) 45 min before	• RMAT total time (s) • RMAT peak time (s)	• ↓ in men • ↔
Tinsley et al. (2017) [35]	Resistance-trained adults Nine Men (20.7 ± 2.8 years) 12 Women (21.5 ± 2.0 years)	Commercially available multi-ingredient pre-workout supplements 4.0 mg/kg for men 3.6 mg/kg for women 30 min before	• Maximal concentric force (N) • Maximal eccentric force (N)	• ↔ • ↔
Jacobson et al. (2018) [37]	Healthy active students 17 Men and 19 Women (19–26 years)	240 mg of Energy drink shot (57 mL) 30 min before	• IFS peak velocity (m/s) • IFS average velocity (m/s) • CMJ power (W) • CMJ peak velocity (m/s)	• ↓ in men • ↓ in men • ↔ • ↔
Chen et al. (2019) [34]	Elite collegiate athletes (tennis, basketball, soccer) 10 Men (21.1 ± 2.1 years) 10 Women (20.4 ± 1.2 years)	6 mg/kg in capsule taken with 500 mL water 24/48 h post-exercise	• MVIC (Nm/kg) • SVIFP (Tlim) (s) • MVIC post Tlim (Nm/kg)	• ↔ • ↔ • ↔

↔ The effect of caffeine supplementation was not statistically different between sexes; ↓ ↑ in males: the effect of caffeine supplementation was statistically different (higher and lower, respectively) in men than in women. 1RM: One maximal repetition; CMJ: countermovement jump; IFS: Isolated forehand stroke; MVIC: Maximal voluntary isometric contractions; RMAT: Repeated Modified Agility Test; SVIFP: Submaximal voluntary isometric fatigue protocol.

**Table 3 nutrients-11-02313-t003:** Summary of studies included in the systematic review that investigated the effect of caffeine ingestion between sexes on the fatigue index.

Author/s	Population	Intervention	Main Outcome Analyzed	Effect on Men vs. Women
Chen et al. (2015) [33]	Elite collegiate athletes (tennis, basketball, soccer) 10 Men (20.10 ± 2.18 years) 10 Women (19.9 ± 0.99 years)	6 mg/kg in capsules taken with 500 mL water 60 min before	• Fatigue index (%)	• ↔
Jebabli et al. (2016) [30]	Healthy active students of Sports Sciences 10 Men (22.9 ± 1.46 years) Eight Women (21.8 ± 0.45 years)	5 mg/kg (Undefined) 45 min before	• RMAT fatigue index (%)	• ↔
Chen et al. (2019) [34]	Elite collegiate athletes (tennis, basketball, soccer) 10 Men (21.1 ± 2.1 years) 10 Women (20.4 ± 1.2 years)	6 mg/kg in capsule taken with 500 mL water 24/48 h post-exercise	• Fatigue index (%)	• ↔

↔ The effect of caffeine supplementation was not statistically different between sexes. RMAT: Repeated Modified Agility Test.

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
