# Peer review of "Effect of Caffeine Supplementation on Sports Performance Based on Differences Between Sexes: A Systematic Review"

_nutrients, 2019, doi:10.3390/nu11102313_

Round 1

Reviewer 1 Report

Replace "gender" with "sex".  Gender is how individuals identify themselves; this is looking at differences between sexes, not genders.  

Regarding formatting: if the article is being published in the United States, many words will need edited for spelling (i.e. "harbour" is spelled "harbor" in the United States; "realisation" would be spelled, "realization", etc.).  I would recommend someone with a very strong background in American English review the paper for misspelled words.

What are the next steps?  What does this lead to?

Did you identify a dosing amount of caffeine that has negative impacts on performance?  If so, you should discuss that. 

Also consider discussing how caffeine is regulated by some governing bodies (i.e. the NCAA--National Collegiate Athletic Association).

Definitely relevant since caffeine is widely used in athletes and non-athletes.

Why were non-exercisers excluded from your review?  Wouldn't they benefit from caffeine supplementation too?

What does the literature say regarding caffeine supplementation in a chronically diseased population?

Author Response

The authors appreciate the time you devoted to reading our manuscript and helping us to craft an improved version of the investigation. We are pleased to clarify your concerns, which we believe have improved the quality and applicability of your work. Please, find below our response to each of your observations. We have made a concerted attempt to systematically address the specific concerns raised for this revision and we have highlighted the alterations to this revision within the manuscript in red for your convenience.

Reviewer(s)' Comments to Author:

REVIEWER: Replace "gender" with "sex".  Gender is how individuals identify themselves; this is looking at differences between sexes, not genders. 

AUTHORS: Thank you for this comment. We have changed “gender” by “sex” throughout the text.

REVIEWER: Regarding formatting: if the article is being published in the United States, many words will need edited for spelling (i.e. "harbour" is spelled "harbor" in the United States; "realisation" would be spelled, "realization", etc.).  I would recommend someone with a very strong background in American English review the paper for misspelled words.

AUTHORS: Thanks for this comment. The previous version of the manuscript was reviewed by a British native. In this new version, we have used US spelling.

REVIEWER: What are the next steps?  What does this lead to?

AUTHORS: Thank you for this comment. In the last paragraph of the discussion, we have included a new section that depicts the next steps for future research.

“Finally, it is of great importance to continue the work in future research on supplementation with caffeine in female populations.  Specifically, it is necessary determine whether women athletes benefit from acute caffeine intake in other forms of exercise or in sports where several physical fitness variables affect overall performance.  In addition, the between-sex difference in the response to other dietary supplements should also be investigated to ascertain whether findings of previous investigations with males are also applicable to female athletes.  Lastly, it is necessary to take into account women’s menstrual cycle when investigating caffeine ergogenicity to determine whether this substance exerts positive performance effects in all phases of the menstrual cycle.”

REVIEWER: Did you identify a dosing amount of caffeine that has negative impacts on performance?  If so, you should discuss that.

AUTHORS: Thank you for your comment. The authors did not identify any dosing amount of caffeine that had negative impacts on performance. In this sense, the authors have added the next sentence in line 316-318: “Interestingly, none of the investigations depicted a negative effect of acute caffeine intake on physical performance.”

REVIEWER: Also consider discussing how caffeine is regulated by some governing bodies (i.e. the NCAA--National Collegiate Athletic Association).

AUTHORS: Thanks, so much for this detail. The authors have included a paragraph in the introduction explaining this (line 61-66): “ Higher doses of caffeine (9-13 mg/kg) do not result in an additional improvement in physical performance [10], while these higher doses might increase the incidence and magnitude of main caffeine-related side effects.  In addition, high doses of caffeine might end in urine caffeine concentrations greater than 15 µg/ml, which is prohibited in the National Collegiate Athletic Association (NCAA) [11].”

REVIEWER: Definitely relevant since caffeine is widely used in athletes and non-athletes.

AUTHORS: Thanks for your interest. Of course, caffeine is one of the most used legal drugs in the world by both athletes and non-athletes. However, the interest of this systematic review is that there are “general” recommendations for caffeine supplementation in athletes that do not specifically differentiate between men and women. These recommendations are based on the majority of cases in studies conducted only in men. Due to the differences between men and women, we wanted to determine if there are differences in the effect of caffeine between both sexes on sports performance. In future studies and / or reviews it should be determined if there are differences in the effect between sexes in other populations such as non-athletes and chronically ill as indicated by the reviewer in other points.

REVIEWER: Why were non-exercisers excluded from your review?  Wouldn't they benefit from caffeine supplementation too?

AUTHORS: Thanks for your comment. Indeed, research indicates that caffeine can benefit non-athletes too. But the interest caffeine intake in non-athletes is often more oriented to compare the physiological and / or psychological benefit of caffeine rather than to sports performance (as is the case of the current analysis). In addition, research has shown that untrained individuals are less reliable in performance measurements than trained individuals.  Thus, the inclusion of both athletes and non-athletes in the same analysis could have biased the results of this investigation.

REVIEWER: What does the literature say regarding caffeine supplementation in a chronically diseased population?

AUTHORS: Thanks for your comment. Your suggestion is quite interesting because it opens the possibility of a future systematic search.  Due to the narrow focus of this investigation -to provide data useful to athletes, we have not included chronically diseased populations, but we keep this comment in mind for future research.”

Reviewer 2 Report

The manuscript responds to a systematic review, focused at comparing the effects of caffeine on sports performance between men and women. It follows the PRISMA guidelines. It describes the procedure followed and identifies the inclusion and exclusion criteria to obtain the final sample of 10 articles.
In the Introduction they provide a background regarding the topic of study. However, it would be advisable to include more references from relevant studies to strengthen it.
The method can be improved. Despite identifying the purpose of the study, the authors don't mention the manuscript's specific objectives. This would enable the manuscript to be understood and easily read in the discussion and conclusions section. In addition, the authors could further specify the process followed to reach a consensus on which studies to include in the systematic review.
The results are presented clearly and concisely. The authors use tables and figures to make reading easier. They also split the results into 4 sections, giving a complete overview of the issue of articles found in the different databases.
Regarding the discussion, the authors include the strengths, weaknesses and future lines of research in this section. It would be more consistent to point this out after the conclusions.
The conclusions are too brief. It is recommended that they expand them according to the sections identified both in the findings and conclusions.
Finally, it is recommended that more recent studies be included in the manuscript (2018-2019).

Author Response

The authors appreciate the time you devoted to reading our manuscript and helping us to craft an improved version of the investigation. We are pleased to clarify your concerns which we believe have improved the quality and applicability of your work. Please, find below our response to each of your observations. We have made a concerted attempt to systematically address the specific concerns raised for this revision and we have highlighted the alterations to this revision within the manuscript in red for your convenience.

Reviewer(s)' Comments to Author:

The manuscript responds to a systematic review, focused at comparing the effects of caffeine on sports performance between men and women. It follows the PRISMA guidelines. It describes the procedure followed and identifies the inclusion and exclusion criteria to obtain the final sample of 10 articles.

REVIEWER: In the Introduction they provide a background regarding the topic of study. However, it would be advisable to include more references from relevant studies to strengthen it.

AUTHORS: Thanks so much for this detail. We have included 3 new references in the introduction section, all published in 2019, in addition to the references included as the result of the response to Reviewer 1’s comments. 

[4] San Juan, A.F.; Lopez-Samanes, A.; Jodra, P.; Valenzuela, P.L.; Rueda, J.; Veiga-Herreros, P.; Perez-Lopez, A.; Dominguez, R. Caffeine Supplementation Improves Anaerobic Performance and Neuromuscular Efficiency and Fatigue in Olympic-Level Boxers. Nutrients 2019, 11, 10.3390/nu11092120.

[7] Durkalec-Michalski, K.; Nowaczyk, P.M.; Glowka, N.; Grygiel, A. Dose-Dependent Effect of Caffeine Supplementation on Judo-Specific Performance and Training Activity: A Randomized Placebo-Controlled Crossover Trial. J. Int. Soc. Sports Nutr. 2019, 16, 38-019-0305-8.

[14] Stojanovic, E.; Stojiljkovic, N.; Scanlan, A.T.; Dalbo, V.J.; Stankovic, R.; Antic, V.; Milanovic, Z. Acute Caffeine Supplementation Promotes Small to Moderate Improvements in Performance Tests Indicative of in-Game Success in Professional Female Basketball Players. Appl. Physiol. Nutr. Metab. 2019, 44, 849-856.

REVIEWER: The method can be improved. Despite identifying the purpose of the study, the authors don't mention the manuscript's specific objectives. This would enable the manuscript to be understood and easily read in the discussion and conclusions section. In addition, the authors could further specify the process followed to reach a consensus on which studies to include in the systematic review.

AUTHORS: Thank you for your comment. We have added in lines 124-128 the specific objectives of the investigation: Specifically, this systematic review focuses on determining the different responses between the sexes to the same caffeine supplementation protocol depending on whether the exercise will be classified as aerobic, anaerobic or when the protocol induced some type of fatigue that could be evaluated (i.e., index of fatigue).”

Besides, we have also improved the explanation of method used to carry out the systematic review.

REVIEWER: The results are presented clearly and concisely. The authors use tables and figures to make reading easier. They also split the results into 4 sections, giving a complete overview of the issue of articles found in the different databases.

AUTHORS: Thanks, so much for your comment. We really appreciate it.

REVIEWER: Regarding the discussion, the authors include the strengths, weaknesses and future lines of research in this section. It would be more consistent to point this out after the conclusions.

AUTHORS: Thank you for your observation. The authors have changed the strengths, weaknesses and future lines of research after conclusions.

REVIEWER: The conclusions are too brief. It is recommended that they expand them according to the sections identified both in the findings and conclusions.

AUTHORS: Thank you for your recommendation. The authors have added a paragraph in conclusion section (lines 468-475): “In summary, 10 studies met the previously defined inclusion/exclusion criteria and were included in this systematic review, aimed to analyze the between-sex differences in the effect of caffeine supplementation on physical/sports performance. The total sample consisted of 221 participants (n=113 males; n=108 females) while caffeine was given by using different sources of caffeine supplementation between 30 and 60 minutes before testing. Studies included in this systematic review measured a large range of variables such as aerobic and anaerobic performance and the fatigue index.”

REVIEWER: Finally, it is recommended that more recent studies be included in the manuscript (2018-2019).

AUTHORS: Thank you for your commentary. Following Reviewer 1 and 2 comments, we have included new references published in 2018 and 2019.
